# Experiences and Suggestions of Nurses Involved in Caring for Migrant Populations in Italy: A Qualitative Study

**DOI:** 10.3390/healthcare12020275

**Published:** 2024-01-22

**Authors:** Claudia Camedda, Maddalena Righi

**Affiliations:** 1IRCCS Azienda Ospedaliero-Universitaria di Bologna, 40138 Bologna, Italy; 2Department of Medical and Surgical Sciences, University of Bologna, 40138 Bologna, Italy; maddalena.righi@studio.unibo.it

**Keywords:** migrants, right to health, transcultural nursing, culturally diverse patients, culturally sensitive care, Italian legislation, Legislative Decree No. 286/98

## Abstract

The aim of this research is to explore nurses’ experiences in caring for migrants, regular and non-regular, within outpatient clinics in Italy. Materials and Methods: Thirteen nurses have been interviewed through online semi-structured interviews, conducted with the support of a questionnaire, made by researchers, consisting of open-ended questions on legislative issues, cultural issues, and regarding best practices. Purposive sampling has been used, along with phone and email recruitment. The audio recordings of the interviews were verbatim transcribed, then examined. This study is a qualitative descriptive fundamental research project. Results: Interviewees highlight migrants’ difficulties in accessing care, critical points of legislation, transcultural skills crucial to nurses, and good practices. A total of 105 labels were developed and grouped into 23 categories under 7 themes (Italian legislation and migrants; structural difficulties in assistance; the influence of politics; the work of NGOs and associations; nursing care; winning strategies; and the role of the Family and Community Nurse). Conclusion: The research highlights how access to care for migrants is hindered by legislative, structural, and cultural barriers, with consequences on the NHS (improper accesses to the emergency room, increased workload, and economic expenditure). For the full realization of the right to health, as enshrined by Art. 32 of the Constitution, changes are needed with respect to regulations and professionals’ training.

## 1. Introduction

### 1.1. Migrants in Italy

The foreign population living in Italy, on 1 January 2021, was about 5.2 million [1], plus about 500,000 irregular foreigners, according to the ISMU Foundation [2]. It is a “young” population [3], with a high rate of unaccompanied foreign minors [4], and the male: female ratio is about 1:1, with wide differences depending on the citizenship of origin [5].

### 1.2. Determinants of Health

According to a 2015 review by the journal *Ethnicity & Health*, worldwide, the barriers to health care for migrants are mainly the right to access in non-universal health care systems and accessibility in universal health care systems, in addition to individual language and cultural barriers; the review proposes to analyze the determinants of access to health services in the corresponding national context [6]. The determinants of health represent an excellent conceptual framework for analyzing the health status of migrants in the Italian context [7].

Migrants live in poor economic and working conditions, resulting in occupational segregation, economic vulnerability, and poverty, both relative and absolute [8]. Economic instability leads families to settle in marginalized neighborhoods where housing quality, safety, and access to public services are low [9]. From an education perspective, 54 percent of foreigners have attained at most a middle school diploma [5], with an impact on health literacy and on approaching the legal and health care systems [9]. Undocumented immigrants tend to underutilize health care systems and delay access to services, mainly due to structural barriers, financial difficulties, lack of health insurance, fear of reporting, and the risk of detention and separation from family; the lack of culturally and linguistically appropriate services further discourages access. The literature highlights the harmful health consequences of lack of access to health care [9].

Compared to Italians, migrants are less likely to go to the emergency department (ED), as happens in other nations [10,11,12,13]; however, they tend to use the ED inappropriately as first access [14,15,16] and/or for non-urgent problems [17,18], probably due to the difficulty in accessing primary and specialty care [19]. Phenomena of racial discrimination limit access to certain types of services, such as preventive health care [6]. Difficulty in accessing care services also results in a higher incidence among the migrant population of ‘preventable diseases’ [20]. To facilitate access to care for the most vulnerable sections of the population, including migrants, several NGOs have activated free clinics on Italian territory where nursing, medical, and specialist assistance is provided. Clinics are financed by private donations or public funding, while the staff is often made up of both paid professionals and volunteers. Although they are not part of the National Health Service, their objective is not to replace the national public system but to attempt to guarantee access to care to those who do not have sufficient information or financial and material resources to go through standard pathways. In fact, the clinics are also useful for those who do not know how to access the National Health Service or do not know their rights; they represent a landmark to obtain information, to know one’s rights, and to renew a prescription, and they correctly direct the population towards the NHS [21,22,23,24]. Some experts illustrate the paradox effect, intended as the phenomenon whereby migrants at the beginning of the migration pathway have a good level of health, which, however, then tends to deteriorate due to a plurality of factors: heavy working conditions (and performance of activities characterized by a higher rate of accidents), precarious housing solutions, relational difficulties, and in general greater social fragility [25]. On the other hand, the health of the migrant population can pose a risk for the hosting population [26], which is why even the United Nations Sustainable Development Goals emphasize that protecting migrants’ health is also crucial for protecting collective health [2].

### 1.3. The Right to Health

From the point of view of law, in accordance with various international treaties and conventions, in Italy health is guaranteed by Art. 32 of the Constitution, which identifies health as a “fundamental right of the individual” and does not bind it to Italian citizenship or residency status, guaranteeing the right to use public health services even to foreigners, regardless of their administrative-legal status. Even Art. 2, referring to inviolable human rights, does not provide for any difference in treatment between regular and non-regular residents, and Art. 3 reaffirms the principle of equality that also applies to foreigners, regardless of the regularity of their residence, in matters of fundamental rights. The constitutional court then added how health is “the foundational core of all other constitutional rights and the inalienable prerequisite for the full realization of the human person” [27] and “therefore must be recognized even for foreigners” [28].

Specifically, for migrants, the reference texts are Law No. 40 of 1998, with the related Legislative Decree No. 286 of the same year and implementing regulations, DRP No. 394 of 1999, which regulate access to care for the foreign population. The National Health Plan 1998–2000 was strongly affected by these regulations and among its various objectives, it aimed to protect the weak and therefore also immigrant foreigners, listing their health among the priorities of the National Health Service (NHS). Circular No. 5 of the Ministry of Health in 2000 made further clarifications on the subject. The 2012 State-Regions Agreement sought to standardize health care for the foreign population on Italian territory.

According to the legislation, mandatory registration in the NHS is provided for migrants legally residing on Italian territory, resulting in them obtaining rights and duties equal to Italians (Art. 34, Legislative Decree No. 286/98). With the registration, a health card is obtained. Foreigners awaiting regularization can be enrolled through a temporary fiscal code, which can be obtained through the Local Health Authority by presenting the receipt of the regularization application. NHS enrollment has the same duration as the residence permit and does not lapse during the renewal phase. Assistance also accrues to dependent family members if they are legal in the territory.

For foreigners not compliant with entry and residence regulations, “urgent or otherwise essential, even if continuous, outpatient and inpatient care for illness and injury is provided in public and accredited facilities, and preventive medicine programs are extended to safeguard individual and collective health” (Art. 35, Legislative Decree No. 286/98). Circular No. 5 of the Ministry of Health of 24 March 2000 specified what is meant by “urgent care” and “essential care”; the determination of urgency and essentiality of services falls within the scope of the physician’s responsibilities [15]. Care is provided through the assignment of an STP code (STP: Straniero Temporaneamente Presente—foreigner temporarily present) for non-EU citizens and ENI code (ENI: Europeo Non Iscritto—European Not Enrolled) for EU citizens, without health coverage in their state of origin, present on Italian territory, but without a residence permit and indigent.

Since 2017, Law No. 47 has provided health care for all foreign minors through mandatory enrollment in the Regional Health System, whether they are children of regular or irregular non-EU immigrants or unaccompanied foreigners.

Of fundamental importance then is the prohibition of reporting, already presented in Legislative Decree No. 286/98 and then reiterated by Circular No. 12, 2019, of the Ministry of Health. The legislator wanted to avoid the situation where the condition of irregularity would result in an obstacle to the provision of therapeutic services: the protection of health is considered a priority over the interest of regularizing one’s stay [29].

The main critical issues in the system are the heterogeneity of regional laws that have produced deep inequalities in the access to care and in the health profile of the migrant population [30]. Likewise, the failure to assign the STP code in some regions is still a big issue [29]. Finally, several studies highlight how health access policies are insufficient unless accompanied by policies aimed at dismantling systemic racism in societies [31]. There is growing evidence that anti-migratory policies are associated with worse health outcomes in immigrant communities [32]. In fact, stereotypes and racism lead to isolation, feelings of rejection, lack of a sense of belonging, distrust in the NHS, and delayed access to care, especially in people with legal status, with worsening clinical status and consequently, higher costs [31]. In addition to these political-administrative aspects, language difficulties and cultural differences highlight how the issue of health promotion towards migrants must be addressed within a broader approach that includes social inclusion in the broadest sense of the term [7].

### 1.4. Nursing Care

Today’s increasingly multiethnic and multicultural Italian society imposes a new challenge on healthcare professionals: to provide culturally competent care which adequately meets foreign people’s health needs [33]. From a nursing perspective, caring for people with cultural backgrounds different from one’s own is complicated due to the cultural connotation of illness, care, the relationship with one’s body, and the manifestation of one’s pain [19].

The question according to some experts is whether it is actually possible to become “culturally competent” simply by studying or experiencing a culture [34,35]. According to these authors, professionals must exercise, rather than “cultural competence”, a “cultural humility”, intended as “a costate commitment to self-assessment and critique, to correct power imbalances in the physician–patient dynamic, and to develop mutually beneficial and non-paternalistic relationships with communities” [34]. In addition, some authors believe that culture-specific training could create stereotypical images pertaining to certain ethnicities and religious denominations, and this could represent a risk in nursing care and therapeutic relationships [36,37]. However, giving individuals from different backgrounds high-quality care is hampered by healthcare providers’ lack of cultural competency [38]. Cultural competence is a complex and multidimensional concept defined as *a set of congruent behaviors, attitudes and policies that come together in a system, agency or between professionals and that allow that system, agency or those professions to work effectively in intercultural situations*; cultural competence allows healthcare professionals to acquire the attitudes, knowledge, and skills to be able to provide high-quality care by also considering people’s cultural background and including beliefs about health and illness, religious influences, main language, values, and other factors that influence people’s health [39].

Le Var [40] identifies the following as major barriers to transcultural care: language and communication difficulties, lack of information provided and shared, services inadequate for users’ needs, misdiagnoses, inappropriate treatment, experiences of racism, lack of equitable access to care services, and consequent poor utilization of services [41]. Shahzad’s review [42], *Challenges and Approaches to transcultural care: An integrative review of nurses’ and nursing students’ experiences*, adds further barriers: inadequate cultural training of staff and difficulties in developing a therapeutic relationship, intrapersonal struggle [43], cultural conflicts [42,44,45], and personal and organizational obstacles. Nurses should develop interpersonal and psychological skills in order to create a therapeutic relationship with patients [42].

In the last few years, some laws have been approved to renew the organization of the Italian national healthcare system; among these innovations, the figure of the Family and Community Nurse (FCN) was introduced in 2020 [46] and was defined as *the professional responsible for nursing processes in the family and community context, […]. He has health as his objective and works by responding to the health needs of the adult and pediatric population of a specific territorial and community area of reference and promoting the health and social integration of services* [47]. Until that moment, in the national territory this figure was not envisaged as an integral part of the national health system, but only implemented through sporadic tests in some rural locations. Remaining an immature professional figure in the Italian context, its role functions are still in the development phase; currently, the FCN is mainly concerned with taking care of the fragile population, chronic patients, people living in urban and sub-urban areas of degradation, and in rural areas poorly provided with services [46].

This study aims to explore the experiences of nurses who provide care to migrant populations, both regular and non-regular, in Italy. The nurses interviewed work within outpatient clinics, dedicated to this segment of the population, run by NGOs and associations of volunteers. It is also intended to investigate the opinion of these professionals regarding the potential role of the Family and Community Nurse (FCN) in ensuring quality care for the migrant population.

## 2. Materials and Methods

### 2.1. Design and Aim

The present study is a fundamental descriptive qualitative. This kind of qualitative design, being a naturalistic inquiry approach, permits a detailed depiction of the phenomenon of interest being studied, influenced by the participant’s experiences and documented by being true to the participant’s words [48]. The aim was to compare the experiences of nurses providing assistance to migrant populations in Italy, analyze challenges related to access to the national health system, legislative issues, necessary transcultural competencies, and successful practices.

### 2.2. Sampling and Partecipants

Sampling was a convenience snowball sampling, which was facilitated through researchers’ personal connections and indirect contacts to associations and organizations. The inclusion criteria were being a nurse and dealing primarily with the care of migrant populations in dedicated out-of-hospital settings, in Italy. The only exclusion criterion was refusal to participate in the study. The recruitment procedure included sending informational modules and obtaining signed consent forms before scheduling interviews; before interviews, the importance of privacy and data treatment was reiterated, and audio consent was requested and obtained before recording. Thirteen nurses from different Italian regions, with different gender, age, and professional experience, were enrolled.

### 2.3. Data Collection

Data were collected through online, individual, semi-structured interviews, supported by a interview guide (Table 1) consisting of open-ended questions. The interview guide, written by the researchers, was divided into three parts: legislative issues, cultural issues, and best practices. The interviews, with an average duration of 30–45 min, were audio-recorded, transcribed and anonymized, and subjected to repeated immersive reading. After every interview, the researcher took field notes that contained preliminary analytical impressions along with pertinent contextual information. There were no repeat interviews.

### 2.4. Data Analysis

Data collection and analysis proceeded simultaneously; analytical notes guided the analysis process and helped to refine subsequent interviews. Then, using a constant comparative analytic approach, differences and similarities within and across data were compared. Interview analysis was carried out using the rapid content analysis method by the two researchers independently; thus, meaning units were isolated and descriptive labels, representative categories or domains, and themes were identified.

Following the performance of 13 interviews, data saturation was reached, as the last 3 interviews confirmed previously identified labels without introducing new or additional ones.

### 2.5. Rigor

To ensure the reliability of the results, detailed records were kept of the entire analysis process, including interview transcripts, field notes, notes on the preparation of the coding process, a list of contextual and analytical themes, and a description of themes.

To secure credibility, at the end of each interview the key topics that emerged were summarized and further comments, suggestions, and additions were collected and integrated.

Confirmability was supported through relevant quotes to illustrate the domains generated, and, in addition, member-checking was performed by summarizing participants’ responses at the end of each interview.

The reporting criteria according to the “Consolidated criteria for reporting qualitative research” (COREQ) checklist were followed.

### 2.6. Research Team and Reflexivity

Personal characteristics—The interviews were conducted by M.R., who graduated in Political Science and International Relations (bachelor’s degree) at the University of Bologna and in Human Rights and Multilevel Governance (master’s degree) at the University of Padua, and student in Nursing during the interviews (now graduated). Data analysis was performed independently by M.R. and C.C. C.C. has a degree in Nursing, a master’s degree in Nursing and Midwifery Sciences, two first-level master’s degrees in Evidence-Based Practice and Clinical Research Methodology, and Management for Coordination Functions in the area of health professions; she is a Professor of Nursing at the University of Bologna and, during the interviews, supervisor of M.R.

Relationship with participants—Since snowball sampling was carried out, only some participants previously knew the interviewer, while the others knew at least one of the previously recruited participants. A relationship was established before carrying out the interviews to share with each participant the objectives of the study, the purpose of the research, the ways in which the data would be reported, elements of privacy regarding the processing of personal data, and other information.

### 2.7. Ethical Aspects

This study was approved by the Bioethics Committee of the University of Bologna (Prot. N. 0150410 del 5 June 2023). Participants involved in this study provided informed consent and consent for the processing of personal data in accordance with national legislation and international guidance.

## 3. Results

Thirteen nurses were recruited; among them, five were men and eight were women, aged 26 to 62 years. The range of years of experience as nurses was between 2 and 42 years and the years of experience in the migration field was between 2 and 12 years. The regions in which they work are as follows: four interviewees in Emilia-Romagna, three in Sicily, two in Trentino, one in Tuscany, one in Lazio, one in Campania, one in Calabria (Table 2).

A total of 105 labels were identified and grouped into 23 categories, which were included in 7 themes. The labels have a frequency of occurrence between 1 and 11. The themes identified are: *Italian legislation and migrants*, which includes 4 categories and 17 labels; *Structural difficulties in caring for migrant patients*, which includes 4 categories and 16 labels flow; *The influence of politics*, which includes 3 categories and 11 labels; *The work of NGOs and associations*, which includes 4 categories and 10 labels; *Nursing care*, which includes 3 categories and 24 labels; *Winning strategies*, which includes 3 categories and 19 labels; and *The role of the Family and Community Nurse* (FCN), which includes 2 categories and 8 labels (Table 3).

### 3.1. Italian Legislation and Migrants

More than 45 percent of the respondents explicitly stated that they see few benefits and instead encounter many obstacles with respect to Italian legislation on immigration and the right to health. In particular, with respect to the latest decrees, many expressed difficulties in showing optimism for the migrant population. Nevertheless, several emphasized the importance, as professionals, of recognizing and valuing the advantages, such as the STP code for undocumented migrants and the exemption (X01) for free access to services, instrumental examinations, and specialist visits. These benefits, however, in addition to being meager, hardly find concrete implementation, as reported by many interviewees, often leaving migrants without health coverage. A quarter of respondents described the Italian legislation as cumbersome, limiting, and dysfunctional and an obstacle to accessing health care. Thus, the right to health care is guaranteed only on paper.


*There are adequate regulations and laws that protect migrant health, such as the Testo Unico for Immigration or Art. 32 of the Constitution, which refers to the individual and not the citizen, resolving the right of health to all. Unfortunately, […] they do not find confirmation and application in reality. Indeed, cases are not uncommon in which many people, especially migrants, find objective difficulties in accessing care.*
(I/7)

Some interviewees defined Art. 32 of the Italian Constitution and universalistic access to care a “utopia”, desirable but unattainable: thus, the principle constitution is missed.


*Art. 32 is a great utopia. Then you need utopias to figure out where to go, but we cannot say that Art. 32 is respected. I don’t see around me access to care universally guaranteed. It would be a big lie to say otherwise.*
(I/2)

Most of the interviewees, when talking about Italian legislation on health and access care for migrants, focused their response not on the right to health care per se, but on the issue of immigration regulation, emphasizing how there is an attempt to complicate, on the one hand, stays for migrants and, on the other hand, sea rescues for NGOs. The goal of institutions and legislation in recent years has been to continuously seek order and rigor, which, however, has often resulted in policies of exclusion. Policies of this kind generate uncertainty in the migrant population with respect to their path to legalization, and expose them to the risk of exclusion, isolation, and crime.


*I met some young people who came to Italy from Pakistan via the Balkan route, they arrived at least two months ago, and the appointment to apply for asylum was given for a month later. Meanwhile, they could not go to the dormitory because they were undocumented, so, they were forced to sleep on the streets. These people, who are left on the streets, end up in the hands of organized crime, drug dealers, etc.*
(I/1)

Below, concrete difficulties in access to care, encountered by the interviewees, are listed. First, as pointed out by many of them:


*the main constraint is that enrollment in the NHS is tied to a residence permit; thus, a person until he obtains his first residence permit, cannot have a treating physician and does not have access to any specialist assessment or examination, except for emergency character.*
(I/3)

Moreover,


*the Cutro Decree, freshly enacted, further exacerbates the difficulties in obtaining a valid residence permit in our country and consequently, complicates everything else. Unfortunately, as we know access to care is tied to obtaining residency, which is linked to having a residence permit and work in order to have a home. In Bologna we see so many people who have jobs, have documents, know the language, but do not have a home, and being homelessness prevents them from registering with the registry office and so, no assistance. For those who do not even have documents, it is even more complicated.*
(I/2)

In addition, obtaining access to care requires documentation that is considered excessive, by a quarter of respondents, which overburdens the work of applicants, as well as wasting time, and waiting times are almost always interminable, as reported by almost half of them. In addition, a further problem is the difficulties in renewing the documents for legal migrants. Finally, the offices are not always efficient:


*The offices, which issue STPs, granted like 5 STPs per week on one day a week. So that is an absolutely dysfunctional organization both from a bureaucratic point of view and from a health point of view because then people do not have the opportunity to receive any kind of care. *
(I/12)

All of the above dynamics, by hindering regular access to care, promote improper access to emergency departments (EDs) and consequently, result in a burden on the public system, as claimed by a quarter of respondents.

### 3.2. Structural Difficulties in Caring for Migrant Patients

Various difficulties related to the operation of the system also emerged during the interviews. The first is the disparity between regions in northern and southern Italy, which also forces many Italians to move to obtain better care. The situation becomes dramatic if we consider that the vast majority of migrants land in the south and the programs of redistribution to different regions do not always work. In addition, the conversion of national health-related norms into regional contexts, borne by the regions, results in different models of care for the migrant population in different regions. Of consequence, nursing care also has to adapt to administrative pressures and organization.


*When [national] norms arrive in the Local Health Authority, they are interpreted and there are possibilities to change situations. For example, an ENI code might have exemptions in one region and not in another. So nursing care, especially territorial care, like the one I do, has to set itself according to the administrative dynamics it goes to meet. If you can have exemptions, you can do things; if you can’t have exemptions, you have to find other avenues, either the resources of the individual or the support of the private social. *
(I/5)

A second difficulty that has emerged is the lack of resources: half of the respondents believe that the resources available within the NHS are limited and insufficient to guarantee access to care for all. Almost half of them say that, even in settings where funding is more generous, the management of resources, already modest, is bad. This results in a lack of universalistic and equitable access to care: in fact, those who end up being excluded from care are migrants.


*Lack of resources is a universal problem, for example, how long do you wait to have a gynecological examination? The discourse of lack of resources overrides the ethnic issue. However, the problem of difficulty of access only affects migrants and is a political will, just as it is a will to keep them in a continuous limbo between legality and illegality. *
(I/2)

The lack of resources concerns material resources, such as beds or dedicated facilities, and organic resources, with very few health workers available.


*One limitation is that there are too few places for people to go. Just think about people who live on the mountains or in the lowlands, if they need to go to the GP and they don’t have the GP, the only place they can go is this [the outpatient clinic] and so maybe they are forced to make even improper access to the PS. In the whole territory of the province of Modena, there is only ours as a direct access outpatient clinic for people who are irregular or not registered with the SSN. It is all volunteer-based, except for the nurse. *
(I/10)

Another nurse states:


*what you see is that definitely the whole hotspot is an overcrowded place. It should have 400 beds, but as a result of so many new arrivals, there are as many as 4000 people. Now the situation has changed a little bit […] But what you observe is that there are 1–2 health professionals for 4000 people and so, basically there are situations that can’t be controlled and so there have been cases of people dying inside the hotspot. *
(I/13)

Third, interviewees described health care personnel as unprepared to assist the migrant population, both culturally and legislatively. Specifically, one-third of respondents complained about the difficulty of interfacing with colleagues who were not up-to-date and educated with respect to current legislation and that therefore hinder access to care.


*The other problem is, if you make the laws then don’t educate the professionals, it’s useless. There has been a new legislation for more than a year that says that minors, children of migrants whether they are documented or not doesn’t matter, are entitled to a Fiscal Code and to assistance. We have experience, until a few months ago, of children who could not do this because, when we showed up at the health services, we were told that they did not knew the law. If you make a law then you don’t do refresher courses and training, it doesn’t work! *
(I/8)

In other cases, operators appear ignorant and uninformed.


*There is a lot of misinformation and ignorance, within the CAS [Centro di Accoglienza Straordinaria: Center of Extraordinary Reception]: the idea is that STP is only entitled to those who have an imminent medical emergency for which, as a result of hospitalization or urgent need to make a visit, then he or she is issued STP. *
(I/12)

To conclude, situations of negligence were also reported with respect to valid health documentation:


*At the time of discharge however I expected a discharge with STP and instead, it was a discharge sheet with last name/“migrant”/born on the day of the visit and a sort of temporary STP code but that had no validity whatsoever, so unusable for a second access. So, from this point of view, absolute negligence regarding the issuance of valid health documentation. *
(I/12)

Finally, the last difficulty that emerged through the interviews concerns the NHS model in general, which appears to be excessively hospital-focused: a model in which health professionals “wait” for patients within hospital facilities. This leads to the risk of exclusion of those with low health literacy, fears, or doubts, as is often the case with the migrant population. In addition, the hospital-focused model drains resources in the territory, which thus remains further deprived. Finally, even within the hospital, adequate services are not always guaranteed for the migrant population, and even when present, they are not always made effectively accessible: many services are considered “on the side” when instead they have a central value in assisting patients with different cultures, such as cultural mediation, which is often absent, unavailable, or even unknown to physicians who are unaware they can activate it.


*Rather than an absence of structural/clinical/specialist resources, perhaps the lack of resources becomes more manifest in “outline” elements, such as mediation, accompaniment, which are extremely scarce and lacking. *
(I/3)

Overall, more than half of the respondents believe that it is necessary to implement a structural change within the NHS and that the distribution of resources should be made more equitable.

### 3.3. The Influence of Politics

A further theme, connected to the first two, concerns the influence of politics. As reported by interviewees, politics intervenes in the distribution of resources, and access to care, from which migrants are often excluded, can be guaranteed if there is a positive political will, as demonstrated by the ad hoc legislation made for the Ukrainian population.

According to interviewees, investments in health care seem to be made more for a matter of image and votes than for their potential impact on people’s lives.


*It is a political idea. If you have a budget with 10 percent for military spending and 2 percent on spending on health and services, you’ve also given yourself an answer. Let’s try to do the reverse, 10% health and 2% armaments, it may be that we already find the solution. We invest in professionals, we train them, we give them tools, and we do ‘low-resource medicine’, meaning an equal allocation of resources and a prudent management, as if it were the management of our house, where you try to implement rewarding paths, even in the paths of logistics and redistribution of resources. You’ll find that you limit waste, that you have more motivated professionals and therefore, everything runs as if it were an organism oriented toward a society that gives a look with respect to the determinants of health. *
(I/5)

Moreover, politics continues to instrumentalize the issue of migration, with propagandistic tones that are not conducive to its proper integration, and above all, as stated by one interviewee, continues to view migration as an emergency and not as a constant and natural human endeavor; hence, the result is a commitment and interest in the issue that is highly dysfunctional in creating appropriate reception models.

Many interviewees glimpsed in their relationship with the Local Health Authority (LHA) of the area where they work an indicator of political will. Interviewees believe that collaboration with the LHA is essential to ensure the effectiveness of the services provided, the exchange of information, sharing protocols, and improving health care for the migrants. A quarter of the respondents reported having a good relationship with the ASL, but another third admitted that they have never perceived concrete collaboration.


*The support was there, but I can say that I felt quite alone. *
(I/11)

For some of them, collaboration with the LHA often proved to be person-dependent. Finally, a quarter of respondents reported the following dynamic: provincial/regional institutions and the LHA tend to ignore grassroots initiatives in the start-up stages of the project, ending up supporting them when they grow and their contribution on the ground becomes relevant and no longer questionable. In any case, even when their support becomes more evident, it still remains insubstantial and hardly materializes in real concrete conventions.

### 3.4. The Work of NGOs and Associations

The situation illustrated above implies the need for NGOs and associations to intervene within the system to guarantee the right to health for the migrant population, as reported by almost a quarter of respondents. NGOs and voluntary associations offer assistance to the migrant population through outpatient clinics, outside the public and hospital systems, managed and financed by private projects.


*Clearly, if the resources were there, there would be no need for the intervention of NGOs. There are not there are resources! Some NGOs participate in projects and access public funds, but they partly finance projects privately because what the public provides is not enough. *
(I/4)

Limited resources, structural difficulties, and the influence of politics result in the shortage or sometimes the total absence of services for migrants, and because of this, 40 percent of respondents define the assistance offered within outpatient clinics as “indispensable.” Indeed,


*if there were no [associations and NGOs], these people would be completely invisible. *
(I/1)

Respondents were asked whether, in their opinion, “popular” clinics increase the interest of policy toward the migrant population, highlighting their needs, or allow, instead, politics to disregard them. Almost half of the respondents admitted that they wonder about this dilemma on a daily basis, but at the same time, none disavowed their work, which allows them to narrow the segment of the population that is excluded from care.


*Partially the risk is taken, and I also believe it is also a moral dilemma. In the sense that, you know it’s a service that should be publicly and nationally guaranteed, and this guarantee is not there, this service is not provided and you go in some way to plug the hole, to make up for the lack of an entity that should instead take charge of this situation. So certainly the risk is to make policy disregard this problem; on the other hand, the risk would be that of standing by and not taking action and not looking for a solution. *
(I/12)

A quarter of respondents explicitly stated that the work of NGOs and associations creates an alibi for politics, which thus feels legitimized to disregard the problem and feels relieved from dealing with the issue.


*I think it’s a big topic that regards all NGOs work that always fills in gaps and provide pretexts for governments and institutions to wash their hands of problems. It certainly happens. I wonder, if there were not all these organizations, would the NHS really be more inclusive? Probably not. *
(I/8)

Not everyone, however, agrees with this idea:


*I don’t think it can diminish political interest about this issue; on the contrary, by working actively they can raise and highlight the difficulties of access to care for this segment of the population, perhaps being able to support in some way the work done by institutions and the NHS. *
(I/7)

Each interviewee sought to provide advice on how best to direct the work of NGOs and associations so as to have a positive impact on the lives of migrants and society. Firstly, more than half of the respondents stressed the importance of documenting their work, collecting and analyzing data, reporting to the authorities the situations encountered, and presenting the real needs of the population. Collecting and analyzing data also makes it possible to create replicable virtuous models.


*The attempt must be to set up virtuous models that set an example and that must become replicable. In any case we would do that [assist the migrant population] because, if we didn’t, people would not know where to go, but in doing it we try to set a model that can be disseminated. *
(I/2)

Other strategies are dialogue tables with institutions, collaborations with the LHA, and a network among various local associations and organizations, to create a “common language” and enable exchanges of information and best practices.

Finally, it remains crucial for almost half of the respondents not to completely replace the NHS, but rather to strengthen the role of territorial medicine, reconnecting with it, for example, redirecting (when possible) patients to their GPs.


*The idea, indeed, is not to replace the NHS, but it should be to try to supplement and make up for the shortcomings, with the idea slowly to make a handover and then leave the project to NHS. *
(I/12)

This is central to the work of NGOs and associations, both to avoid the risk of total disinterestedness on the part of the public system and politics, and also because


*you cannot save people alone, you have to collaborate, and this is something that as a health professional doesn’t immediately come to mind, we don’t see ourselves in the role of policies creators, however, we are and we have the responsibility to collaborate, to be heard, be seen, to get the best care possible and to make those responsible for public health to really take charge of all patients. *
(I/3)

Indeed, a big problem is that often these clinic projects are term-limited projects which means they cannot guarantee the continuity of care: reconnecting with the NHS is then the most effective strategy.

### 3.5. Nursing Care

The interviewed nurses provide care primarily to adult men; few have an almost exclusively female or mixed clientele. The migrants assisted have diverse origins, mainly Bangladesh, Pakistan, Eastern Europe, and North and Central Africa (Morocco, Tunisia, Senegal, and several sub-Saharan countries).

Only three respondents stated that they did not encounter any cultural barriers, but they also stated that they had worked several years abroad, always in contact with foreign populations; these experiences have probably provided them with the tools to approach other cultures more easily. One of them also admitted that in terms of the actual delivery of care,


*in my opinion we have many more prejudices than what the actual situation is. *
(I/12)

The main difficulty encountered by more than 50 percent of the respondents, in providing assistance to people with cultural backgrounds different from their own, is the language.


*You get used to it, some French, English, Google translator, however clearly there is a limitation for you that then have to find an answer to the need, but it’s a limit especially for the person himself to express his need. *
(I/2)

Some respondents said they encounter ethnographic/anthropological barriers.


*The typical example is trying to treat an animistic epileptic: try to give him medicine when he thinks he has a demon in him and needs to go home to get the amulet. The issue is cultural and is based on the different approach to illness from people to people. *
(I/2)

The differences may be in the way pain is expressed, the meaning given to the illness, or in the explanations given during a medical history.


*I remind of some interviews, at first visits, in family history, “My father died when he was about 40 years old,” “How come? Do you remember? Did he have a particular pathology?”, “Mmm, he was a healer and had accumulated too many evil spirits.” *
(I/3)

One-third of the respondents, then, experienced a misuse of drugs by migrants or in general non-adherence to therapeutic prescriptions. One respondent states:


*there is a lot of misuse especially of antibiotics. There is no knowledge of the function of the antibiotic which therefore, is used for everything and there is an immoderate demand and use. Almost everyone travels with antibiotic in their backpack without knowing the function. Often then there’s abuse: you give a medicine, you explain it, there’s understanding on the part of the patient, however then, “the toothache didn’t get over it, so I took the whole blister pack.” *
(I/12)

Some migrants find it difficult to understand some aspects, such as, for example, prevention, which they perceive as distant from their own culture.

Additional aspects to be taken into account in the nursing care of migrants, according to the interviewees, are direct contact and gender relations. The latter can create difficulties between professionals and patients of different genders: some respondents said they felt unrecognized in their role as nurses by male patients, while others described situations in which Muslim women refused to be examined by male doctors. In some cases, however, difficulties inherent in gender relations also emerge within the households themselves, where it is difficult to enforce the privacy of the wife.

In addition to these cultural difficulties, there are also difficulties related to the identification of the rights of the assisted (which depend on their status and therefore on the code assigned to them): these difficulties relate both to the interfacing with the system by migrants, who do not always possess the right information, and to the interviewees’ “cultural” preparedness.

Respondents were then asked whether they find migrants more distrustful or trusting toward our NHS. Five respondents stated that they mainly encounter mistrust due to “cultural distance.” Three respondents, however, emphasized that mistrust is more toward the Western model of medicine than toward the NHS itself, more toward the hospital than toward outpatient clinics.


*Moreover, distrust seems to depend not so much on the NHS nor on the Western model of medicine, but on the bureaucracy, which makes migrants exhausted and perplexed. *
(I/4)

One of these five interviewees, however, pointed out that actually distrust is only initial and that it is then overcome through integration with our culture.


*Of course, it is different for those who arrived 40 years ago and have settled here; for them, even the Western medicine has been assimilated. For those who have just arrived, the distrust is enormous. *
(I/2)

In total, therefore, nine respondents said they experienced more trust than distrust. And only a quarter stated that they found more distrust in women than in men (it must be considered, however, that most of the respondents have a male-only clientele and thus were unable to answer questions about male–female differences, due to a lack of information).

Almost all respondents, 11 out of 13, stressed the need to integrate the nursing degree programs, especially given the professional evolution and the changing demographics of the population nowadays in Italy.


*In a totally globalized world, huge masses of people move by plane or on foot and it is desirable to include in academic courses, elements of medical anthropology, ethnopsychology, ethnopsychiatry—this would be the minimum. It is important for a future physician, to know about human trafficking, risks, prostitution, etc. this allows you to understand many things and to avoid asking questions that risk bringing back bad memories and souring the relationship of trust as well. You need a lot of ECTS on these topics! *
(I/2)

During the interviews, however, it was difficult to identify which aspects could concretely be incorporated within degree programs. Suggested topics included how to approach a patient with a different cultural background and language skills. However, in general, respondents almost all seem to agree that there is no need for overly detailed or in-depth courses, but generic notions suffice.


*Teaching should not be a “I teach you in this culture how things work” also because it would never be enough. It is true that we can make a prediction of the major communities present in Italy. However, there will always be someone from a small town in Mali that doesn’t fit into your knowledge. It is more a matter of approach, making it clear what is an open approach. *
(I/8)

According to nine of them, knowing patients’ culture is the key to providing quality care, in addition to the fact that it encourages the approach of the migrants themselves.


*As nurses we care for the whole person so we cannot disregard the culture—obviously getting help from the cultural mediators. Certainly, knowing the cultural factor well allows us to achieve certain results at the nursing level. *
(I/4)

Interviewees also agreed on including courses on current events and topical issues within primary, middle, and high school education to promote knowledge of the phenomenon of migration to all citizens. A greater knowledge would help to break down prejudices, as claimed by five respondents. Only one respondent raised the doubt, in line with the scientific studies cited in the introduction to this paper, that courses on other cultures risk creating stereotypes. This risk was not perceived by almost half of the other respondents, because


*our universities are based on scientific methods so the courses introduced must be proven, this limits the risk. The more you know, the less stupid and simple answers you give. The more you articulate knowledge, the more articulate your answers. *
(I/2)

### 3.6. Winning Strategies

Respondents were asked to share the winning strategies identified within their clinics, in order to facilitate the spread of these best practices among professionals from different regions. Some have already been illustrated within the themes “The work of NGOs and associations” and “Nursing care”; others have been divided into three categories concerning the team, the patient and more structural aspects.

Regarding the team, nine out of thirteen respondents believe it is a successful strategy to involve a cultural mediator, who is not just a translator, but rather helps the professionals in approaching the patient and leads the patient to have more trust.


*The presence of the mediator is crucial in breaking down most of the cultural barriers that might come up. In the sense that when you have created communication, you have created trust and automatically you have created an opportunity to interact, to explain, to act consensually on both sides and with the specific that we never talk about an interpreter but a cultural mediator. It is not a person who simply translates from my language to yours and vice versa, but is a person who creates a channel of trust, a channel of interpretation of the message, who knows your culture and knows how to get the message across in the most understandable way. *
(I/12)

Only one respondent, while acknowledging its potential, pointed out the risks of the cultural mediator, who can become an obstacle if the mediator himself is prejudiced, as supported by an incident that occurred between Afghan teenage girls and an elderly mediator, during a gynecological visit: the mediator’s assumption of the girls’ virginity (as they were not yet married) prevented them from expressing themselves freely.

Referring to the team, other suggestions are multidisciplinary teams, sharing among colleagues after each intervention/experience, and ongoing training and updates, to know the legislation, to deepen cultural issues, and to share strategies identified.

Secondly, interviewees listed winning strategies in approaching patients. Actually, this strategies could be grouped simply under the category of “acting as professionals”. Indeed, the main strategies are to establish a relationship of trust and “hook” the person, devote time, explain treatments and ask for feedback, dialogue with the patient especially when the patient expresses a need that is not purely physical, have respect and empathy and approach without prejudice, do not get too close or touch the person, and ask for consent before each procedure. One respondent suggests learning even just a few words in the patient’s language to encourage closeness. In addition, always try to give autonomy to the person and maintain contact.


*The strategy that works the most is first of all to give autonomy: whatever you want to do, if you don’t make the person a protagonist in his/her own journey, there will always be a problem. If you don’t give autonomy and don’t make him/her understand what the treatment is, what hospitals are of reference, the fact that, if he does not learn the Italian language there will always be a problem and he will always be a step behind, even for the job search, when you tell him that with a better workplace he will also have a better chance of getting treatment and getting out of the dynamics of “Caporalato” and everything else… so, at the center is a person with a project migration. All the rewarding strategies… what goes into increasing the level of autonomy, awareness and interrelationship between the person and his or her environment is something that is rewarding, whether it is an Italian course or going alone to the doctor or choosing the doctor after qualifying, suing an employer who doesn’t pay… start to use the tools, the few that the state gives, to be a full citizen, this is a winning strategy. *
(I/5)

According to two interviewees then, it is crucial to offer assistance, not only physical/clinical but also psychological, given the path that these migrants are forced to experience and overcome.

In addition, interviewees emphasized the value of raising awareness among the community and health workers, on the one hand, and on the other, offering information and guidance to migrants, who often do not know how to approach our NHS.

Two interviewees stressed the importance of documenting their work, as already happens inside the hospital. Documentation protects the practitioner and the caregiver and, moreover, if performed using, for example, the electronic file can further facilitate communication and sharing among professionals.


*There is a tendency not to record, not to write … this makes it more difficult to work, it makes save 3 min not to record an access to the clinic, but then the next time you don’t know what you did or what your colleague did… again we are talking about a logic of safety for us and for the patients. Already they are people who are used to “not existing” on paper, not recording, documenting, writing, providing a piece of paper back after a visit, always seems to me to be neglecting people who have little opportunity to act in their turn. *
(I/3)

Finally, an interviewee working in an emergency context (at the landings), proposed the drafting of clearer protocols and guidelines on how to move in emergency contexts.

In general, a third of respondents mentioned that the real winning strategy remains having a lot of elasticity/resilience.

### 3.7. The Role of the Family and Community Nurse

The last question in the questionnaire intended to explore interviewees’ opinions regarding the figure of the Family and Community Nurse (FCN) and its possible future role in ensuring quality care for the migrant population.

Eleven out of thirteen respondents believe that the FCN can assume a relevant role in migrant care, fill a void present in our NHS, and overcome the fragmented nature of the system itself: indeed, migrants who approach the NHS must go to different counters, often ending up stranded in bureaucratic quibbles.

The FCN could provide continuity in the care pathway and could bring back the responsibility of care within the NHS.

Only one respondent presented a doubt: the potential of the FCN’s role changes based on the type of migrant population being considered, the settled and the transient.


*By settled population I am talking about a population that is assimilated to the Italian population, so, I wonder and ask you: if community nursing is useful for me, it should be useful for my Senegalese neighbor as well, right? There should be no difference between me and him, so yes, for the settled population absolutely yes, but not so much specifically for the migrant person, but as a useful service for all citizens and the whole population present throughout the territory. If, on the other hand, we are talking about the transient population, then it becomes a little bit more complex, how do I access the FCN service? Do I access it through an STP code? How long am I going to stay in the territory? […] Probably for this type of population you ask for a more specific type of service, more focused on what is the trend and transit in Italy. So maybe not, the FCN would not be the ideal solution. *
(I/12)

All interviewees, while seeing the potential of the FCN figure, agreed that there are still many shortcomings and that this figure, nowadays, is not yet ready to offer quality care to the migrant population. According to two interviewees, there is a lack of willingness and interest to reach all segments of the population; others agree that there is instead a lack of specific training, which for now is more focused on other segments of the population. Finally, what is lacking according to two respondents are, once again, resources.


*For example, there are many FCN in suburban neighborhoods in Bologna, but there are big obstacles, so people end up in our outpatient clinic, because FCN does not have the elements, the resources. There must be created special systems to be able to best ensure continuity of care, etc.! *
(I/3)

## 4. Discussion

The research shows how the migrant population, especially if irregular, is often excluded from official health care services; this is obviously not a purely Italian phenomenon, but occurs in several European countries, despite the presence of conventions that oblige the EU members to uphold the human right to health [26]. The barriers to access are of two types: on the one hand, there are structural barriers conditioned by legislation, on the other, there are barriers dictated by the reduced capacity of health services to meet the needs of the migrant population [6].

Generating social and institutional changes that remove barriers to access to care is fundamental to ensuring health for all [6]. The need for changes is not determined by morality, but is an economic and security issue; trivially, migrants’ health influences the health of Italian citizens themselves and as also promoted by the United Nations, migrants’ health should be protected “for the realization of their life projects, but also for greater protection of the collective health of the host community” [2]. On the other hand, since health care is often sadly reduced to an economic issue, the exclusion of migrants from services, such as prevention and health promotion, leads to an increase in improper admissions to emergency rooms, resulting in a heavier workload and increased economic expenditure. Finally, moving beyond economic issues, migrants’ exclusion from care is contrary to the constitutional principle of the right to health, enshrined in Art. 32 of the Constitution. From this perspective, the goal, from a political, social, and economic point of view, should be to avoid the paradox effect, mentioned in the introduction, by also expanding services (especially prevention and health promotion) to migrants. Despite this, there seems to be a lack of political interest, which, however, is opposite to the very nature of our NHS: as a universalistic system, the Italian health system cannot allow the exclusion of such a sharp slice of the population.

Legislative Decree No. 286/98 and its implementing regulation, DPR No. 394/99, recognized the right to health to foreigners, both regular and not, and promoted their inclusion in the NHS [30]. However, after more than twenty years, there are still numerous difficulties met by health professionals in providing care to this segment of the population; these are to be attributable in part to the ignorance and incompetence of health professionals regarding the law, in part to the dysfunctionality of the counters and the system in general.

Moreover, the heterogeneity of regional laws has generated profound inequalities in services access and in health profiles of the immigrant population [30].

Nurses, on the one hand, have the duty to broaden their knowledge with respect to the rights of their patients, because while practicing the profession they also have an advocacy task towards people; on the other hand, they should deepen their cultural competences in order to meet and satisfy the needs of all segments of the population.

The interviewed nurses reiterated the need to integrate nursing courses with generic notions of medical anthropology, ethnopsychology, ethnopsychiatry, teachings on how to approach different cultures, and language skills. This need is reported in numerous studies [41,42,49,50]; in particular, Vitale [50] proposes to integrate nursing courses with teachings about cross-cultural nursing models to enable the understanding of other cultures’ values and develop appropriate nursing techniques. The need for the expansion and integration of degree programs stems from the fact that nowadays, hospital users often include foreign patients: new skills and knowledge are then needed both to ensure quality care, but also for the health workers themselves, to reduce their difficulties in approaching and creating a therapeutic relationship proper to the nursing profession. Despite many studies supporting this need for integration, despite the evolution of the profession and the demographic transformation, nursing courses have not yet been updated in this regard [50].

Even today, despite the changing demographics, there is little attention to multiculturalism in the university context. Just think of the ECTS devoted to language skills within the three-year course of study. Respondents identified communication and the language barrier as their first difficulty: this is also confirmed by Vitale in the study *Transcultural health: attitudes, perceptions, knowledge of Italian nurses. An observational study*, in which 41.30% of respondents cited language as the main cultural barrier [50]. Therefore, it is possible to assume that the introduction of language courses could be the first and easiest step to update nursing courses.

Vitale’s study is also in line with this research with regard to the other difficulties encountered by the interviewees: the lack of knowledge about access to health services by the migrant population, the rejection of therapies and procedures perceived as culturally distant, and the obstacles imposed by religion [50]. These difficulties highlight how in the management of the migrant patient, in addition to the basic nursing skills, the professional needs advanced skills, such as the ability to deal with cultural mediators and anthropologists, the understanding of relevant legislation, the knowledge of hospital and territorial services, and a mastery of foreign languages; all these contents should be ensured through updating and training [51]. Once again, then, the problem could find at least a partial solution in nurses’ training.

Furthermore, in agreement with this research, Vitale’s study confirms the absence/shortage services adequate for migrant people and adds that this is probably the true face of discrimination [50]. Even just the fact that Vitale’s 2002 study and this research, conducted in the current year, 2023, identify essentially the same difficulties in providing care to patients with migrant backgrounds should suggest a reflection on the efforts and investments made over the decades in health care for migrants. Looking at the concepts set out in the introduction, the lack of efforts and investments becomes even more evident: indeed, the NHS shortcomings, regarding foreigners’ health, illustrated at the Conference of the Italian Society of Hygiene (2006), by the then-Minister of Health Livia Turco, remain relevant even today (“cultural approach, […], humanization in the relationship services-immigrants, […] cultural mediation between public health services and populations immigrants, […] communication about rights, […] and preventive services”) [30]. Thus, there is still a long way to go for the realization of the right to health, as enshrined in our Constitution, a right for all: the goal of NSP ‘98-00 to protect the weak and therefore also immigrant foreigners has still not been achieved after twenty-five years; likewise, also the political will of Legislative Decree No. 286/98 to guarantee irregular migrants access to care “not only essential and urgent, but also continuous and preventive”, has not yet been realized.

A possible solution to this situation, which could embrace health policy and care, the determinants of health, and migrants’ involvement, is the figure of the Family and Community Nurse, as stated by almost all of the interviewees. Specific training and refresher courses would enable the FCN to acquire the necessary skills to provide quality care; however, in addition to investments in training, investments should also be made in the resources available for the territory and primary care. In any case, the FCN could be more of a solution for the settled migrant population, considered an integral part of the target community in which the FCN acts, than for the transiting one.

However, given the current political mood on the issue of migration, it is difficult to be optimistic about health developments for migrants, as also stated by some interviewees. In particular, regarding the topic of investments, the interviewees emphasized the necessity to take into account all the determinants of health in order to create a fertile territory where the right to health can flourish. Today, however, the political climate is rather hostile, and anti-migration propaganda seems to be the government’s priority.

Beyond politics, however, it is desirable that nurses always fight to ensure that the right to health which in our Constitution is universalistic and in our Code of Ethics should be “without any discrimination”.


*We do not think to exaggerate by saying that anyone who has had, at least in a professional sphere, encounters with an immigrant or, rather, with multiple immigrant citizens, has wondered, had curiosity, came up against the evidence of a bureaucracy that tends to exclude otherness, in some cases has felt powerless on the relational or clinical, sometimes outraged by reactions and attitudes. Nothing new for those who have chosen to work in helping relationships. But what is new in the relationship with the foreigner, often the realm of shared prejudices, more or less conscious, is that we have measured, and we measure with concrete hand our powerlessness: the communicative linguistic, relational, political-organizational, cultural therapeutic one. Again, there is the awareness that in order to overcome this impotence, we must cross the diaphragm that separates us from each other and share information, impressions, discoveries, strategies. In the mirror-function that immigration forces us, we find our weaknesses but also the motivations to be protagonists of what, with good reason, we can call real pathways to public health and the right to health “without any exclusion”.*
[30]

## 5. Conclusions

This research has shown that the road to realizing the right to health for all is still a long way off. Access to care for the migrant population is hindered by two types of barriers: structural and legislative barriers and cultural barriers. This lack has consequences for the NHS (encouraging improper access in emergency rooms, increasing the burden workload and health economic expenditure), and on the health of Italian citizens themselves, as also stressed by the SGDs of the United Nations. Moreover, there are an ethical–moral issue to deal with and the contradiction with Art. 32 of the Constitution and the universalistic health care system. In this sense, therefore, there are two fronts on which to act: on the one hand, legislation and its full implementation, and on the other, the training and updating of health personnel.

Despite the many obstacles, interviewees also highlighted the advantages present in the current legislation and shared good practices, already experienced within their clinics, with the aim of also promoting quality care for the migrant population.

Finally, the new figure of the FCN, if properly trained, could be a (partial) solution to the problem.

## Figures and Tables

**Table 1 healthcare-12-00275-t001:** Interview guide.

**1. Questions about Legislation**
Which advantages and obstacles regarding the current Italian immigration legislation you find in providing assistance to migrants, regular and undocumented? Are the resources, made available by the NHS (National Health System), sufficient to guarantee to everyone the Right to Health, enshrined in Article 32 of our Italian Constitution? Please argue your answer. Do you think that the work of NGOs and Associations, in offering health care to migrants, risks to diminish political interest towards this segment of the population or on the contrary, it may strengthen NHS’s responsibility and taking in charge? With reference to this issue, how would you define the collaboration with the Local Health Authority (LHA) within the territory in which you operate?
**2. Questions about Cultures**
In providing assistance to the migrant population, do you encounter any particular cultural difficulties? If so, what are they? Is there a community with respect to which cultural barriers are most evident? In the assisted migrant population, do you find distrust or trust in the Italian health care system? In your experience, is there more distrust in the male or female population? What should be the main knowledge and skills of professional healthcare workers working in cross-cultural settings? Language skills, interpersonal skills, etc.? Do you think there is a need to supplement nursing degree programs with such knowledge and skills? With respect to this issue, do you think it is possible and beneficial to disseminate specific knowledge with respect to specific cultures, or do you feel that this could then lead to biases in approaching individual patients?
**3. Good practices and suggestions**
What are the winning strategies that you have found within your project, thanks to your experience, and that you recommend implementing in providing health care to the migrant population, regular and undocumented? Do you think that the emerging figure of the Family and Community Nurse can represent an opportunity to ensure quality care for the migrant population, both regular and undocumented, in Italy? If yes, how?

**Table 2 healthcare-12-00275-t002:** Characteristics of participants.

Interviewee	Sex	Age	Degree(s)	Years of Experience as a Nurse	Years of Experience in the Migration Field	Region of Practice	Experience Abroad as a Nurse	Foreign Languages Spoken
I/1	F	31	Bachelor’s degree in Nursing Master’s degree in Coordination for Health Professions	8	4	Tuscany	Oxford, UK (1 year)	English Spanish
I/2	M	35	Bachelor’s degree in Nursing Master’s degree in Scrub Nursing	10	10	Emilia-Romagna	Balkan route (several times) Mediterranean various SAR zones (several times)	Spanish (first language) English
I/3	F	27	Bachelor’s degree in Political Science Bachelor’s degree in Nursing Master’s degree in Evidence-Based Practice Master’s degree in Biostatistics	2	2	Emilia-Romagna	None	English French
I/4	M	55	Bachelor’s degree in Nursing Master’s degree in Critical Care Area Postgraduate course in Tropical Medicine	30	11	Trentino	Libya (1 year) Egypt (1 year) Tunisia (1 year) Iraq (3 years) Kurdistan (3 years) Afghanistan (2 years)	English French
I/5	M	40	Bachelor’s degree in Nursing Master’s degree in Coordination for Health Professions Master’s degree in Family and Community Nursing	20	10	Campania	Moldova (1 month) Central Mediterranean various SAR zones (4 months)	English French
I/6	M	62	Bachelor’s degree in Nursing	42	12	Sicily	Albania (10 days) for Kosovar migrant emergency	None
I/7	F	48	Paediatric Nursing degree Master’s degree in E-teaching and Tutorial Teaching Postgraduate course in Tropical Medicine and Health Cooperation Bachelor’s degree in Political Science for International Relations Master’s degree in Coordination for Health Professions (ongoing) Several courses in Humanitarian Emergency Management and Project Management	30	5	Trentino	Iraqi Kurdistan (1 1/2 years) Sudan (1 year) Afghanistan (3 months) Iraq (3 months) Cambodia (2 months) Somalia (2 months) Ukraine (2 months)	English
I/8	F	45	Bacherlor’s degree in Nursing Master’s degree in Islamic Studies	6	4	Lazio	UK (4 years)	English Spanish French Arabic Turkish
I/9	F	45	Bacherlor’s degree in Nursing	22	3	Sicily	None	English
I/10	F	30	Bacherlor’s degree in Nursing	7	2	Emilia-Romagna	None	English
I/11	M	39	Bachelor’s degree in Nursing Bachelor’s degree in Social Work Education Master’s degree in Family and Community Nursing Master’s degree in Coordination for Health Professions (ongoing)	6	2	Emilia-Romagna	Watford, UK (2 weeks)	English Spanish
I/12	F	29	Bachelor’s degree in Nursing	7	2	Calabria	Tanzania (1 month) Madagascar (9 months)	English Spanish
I/13	F	26	Bachelor’s degree in Nursing Master’s degree in International Cooperation and Educational Inclusion	5	2	Sicily	Bosnia–Croatia border (1 month) South Sudan (3 months)	English

**Table 3 healthcare-12-00275-t003:** Thematic analysis.

Themes	Subthemes	Codes
italian legislation and migrants	interviewees’ impressions and opinions	Struggle to be optimistic
		See few advantages and many obstacles
		Recognize benefits: STP, access to exemptions, primary care, access to counseling, etc.
		Be a maze/limiting/dysfunctional
	rights guaranteed only on paper	Guarantee, on paper, the right to health and the access to care
		Not find application in reality
		Not comply with the constitutional principle
	exclusions and consequences	Exclude people
		Seek order and rigor but achieving the opposite
		Increase the risk of crime
		Increase uncertainty
		Make complicated to stay/rescue at sea
	practical problems	Require excessive documentation/numerous prerequisites
		Have long waiting times
		Struggle to renew documents
		Tie NHS enrollment to residence permit
		Burden the public system/increase improper accesses in ER
structural difficulties	regional differences	Convert national norms into regional contexts
		Adapt to administrative/organizational dynamics
		Have disparities between North and South
	unprepared staff	(Not) be up-to-date and educated with respect to legislation/Be ignorant and uninformed/Be negligent with respect to issuing valid health documentation
	lack of resources	Have limited and insufficient resources for all
		Disadvantageously manage the little resources available
		Not guarantee universal and equitable access to care
		Force to use private project funds
		Not have sufficient beds or dedicated facilities
		Have availability of few healthcare workers compared to the large migrant population
	hospital model	Have a hospital-centric approach/Wait for patients in the hospital/Not have resources for the territory
		Ensure health services but not care about making a service effectively accessible/Not guarantee “side” but necessary services (e.g., cultural mediator within the hospital)
		Exclude those with poor health literacy
		Be there need for structural change
		Be there need to make the distribution of resources more equitable
		Be there need to consider all determinants of health
the influence of politics	allocation of resources	Ensure access to care when there is willingness (see Ukrainians’ ad hoc legislation or individuals pathways such as Asylum Seeker and Refugee Protection Service)/Allocate resources according to policy idea
		Allocate resources unequally
		Make policy choices that put lives at risk/ Choose where to invest (military spending rather than health spending)
		Choose investments for image and votes
	instrumentalization	Use the issue of migration in an instrumental and propagandistic way
		View migration as an emergency and not as a constant and natural human initiative
	lha (local health authority)	Not perceive concrete collaboration
		Have a good collaboration
		Ignore people’s initiatives initially then supporting them when they become relevant and strong
		Never sign concrete conventions/offer little concrete support
		Have person-dependent collaborations
the work of ngos and associations	popular outpatient clinics	Offer services that are not present
		Assist people who are otherwise “invisible”
	moral dilemma	Wonder the moral dilemma/Have the doubt
	relationship with politics	Create an alibi/Provide excuses to disregard the problem
		Lighten the politics
	how to act	Document, analyze and denounce
		Create tables of dialogue with institutions
		Create a network between associations and collaborations with ASL
		Create virtuous and replicable models
		Strengthen the role of territorial medicine/Reconnect with territorial medicine/Not replace/ Redirect pts to their GPs
nursing	difficulties in assistance	Meet no barriers
		Communicate in different languages
		Manifest pain
		Give different meaning to illness/ Provide culturally-influenced explanations
		Use medications incorrectly/excessively/ Not adhere correctly to therapy
		Encounter ethnographic/anthropological barriers
		Not understand aspects distant to one’s own culture, such as prevention
		Deal with patient contact
		Be a woman/Gender relations
		Identify/understand rights
		Interface with the system
		Train the staff
	migrant patients’ trust and distrust	Be wary initially
		Be distrustful due to “cultural distance”
		Be wary of the (western) model of medicine more than of NHS/Be more wary of the hospital than of the outpatient clinic (therapies)
		Be more distrustful as women
		Have trust
		Lose trust because of bureaucracy
	how to improve	Know the culture to provide quality care/favor migrants’ approaching
		Embed generic notions (elements of medical anthropology, ethnopsychology, ethnopsychiatry) within the degree program
		Break down prejudices with knowledge
		Structure new cross-cultural courses in a scientific manner
		Incorporating language skills
		Act as professionals
winning strategies	within the staff	Involve the cultural mediator (help with language but also with approach)
		Share among colleagues after each intervention/experience
		Multidisciplinary teams
		Train
	with the patient	Establish a relationship of trust/’Attach the person’
		Dedicate time/Ask and explain treatment/Dialogue with the patient
		Have respect
		Don’t get too close, don’t touch the person
		Ask for consent before each procedure
		Have no judgment
		Give autonomy to the person
		Offer psychological assistance
		Maintain contact
		Have elasticity
	structural	Raise awareness in professionals and give information to migrants
		Collaborate with LHA
		Document/share among professionals a computerized electronic record
		Overcome the dichotomy between outpatient clinics for people who are enrolled in the NHS and outpatient clinics for people who are not enrolled, with STPs
		(in the most emergency contexts) Create an operational structure with protocols and lines on how to move
the family and community nurse (fcn)	Potentialities Of the facn	Have a relevant role
		Be able to fill the gap/Overcome the fragmented nature of the system
		Provide continuity
		Bring the responsibility of taking in charge back within the NHS
		Depend on the type of migrant population: useful service for settled citizens and migrants—not a solution for the transiting population
	What is still missing	Have an interest in reaching the entire population
		Specific training
		Resources

## Data Availability

Data are contained within the article.

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
