# Peer review of "Experiences and Suggestions of Nurses Involved in Caring for Migrant Populations in Italy: A Qualitative Study"

_healthcare, 2024, doi:10.3390/healthcare12020275_

Round 1
Reviewer 1 Report
Comments and Suggestions for Authors
Comments to the Author
Thank you for the opportunity to review this manuscript.
Your topic is very interesting. This makes a relevant contribution to the field. I suggest the following changes/considerations to improve your work. Please find my feedback below:
- According to the COREQ checklist, Domain 1: “Research team and reflexivity” parts need to support. Please add more information about this.
In general, the “materials and methods session” need reinforcement.
- Line 174-
Were transcripts returned to participants for comment (to secure credibility)?
- How many researchers participated in data analysis? This needs to be presented in the data analysis section.
- Line 185 –
Was each interview conducted with each research participant only once? Were there no repeat interviews?
- Add the research questions you used during the interview.
- Please describe the research design and data analysis in more detail.
Author Response
Dear reviewer,
I thank you for the considerations indicated, which were essential for improving our manuscript. I'll show you below how they were integrated; I hope you can provide us with further considerations for improvement if necessary:
- The parts of Area 1 of the COREQ checklist: "Research and reflexivity group" have been included in the "materials and methods" chapter in a specific section.
- The entire "materials and methods" chapter has been radically modified; further details on the methodology used have been included and, furthermore, it has been divided into subchapters for greater clarity.
- in the sub-chapter "rigor" we reported "To secure credibility, at the end of each interview the key topics that emerged were summarized and further comments, suggestions and additions were collected and integrated."
- we indicated the number of researchers who participated in the data analysis in the "Data analysis" subchapter
- we indicated that each interview was conducted with each research participant only once and there were no repeated interviews in the subchapter "Data collection"
- We have added the research questions in Table 1.
We remain available for further changes.
Best regards.
Reviewer 2 Report
Comments and Suggestions for Authors
The aim of this study is to identify nurses' experiences of caring for migrants with both regular and irregular administrative status in outpatient clinics in Italy.
I congratulate the authors for this excellent initiative, which will contribute to a significant improvement in the quality of health care for all people, not only those from countries other than the one in which they now reside.
I will now set out my suggestions for improvement in each of the relevant sections.
INTRODUCTION
The introduction contains information that is useful for contextualising the study and highlighting its origins.
However, there are several aspects that deserve to be reconsidered and reformulated.
For example, statements such as 'migrants tend to use health services more and inappropriately' are questionable. Only one study is cited as saying this, but there is a large body of scientific literature to the contrary. I suggest that this aspect be investigated.
It is also pointed out that the national legal framework does not distinguish between citizens and non-citizens. The very fact that people in an irregular administrative situation are referred to as 'non-citizens' is a violation of human rights. I would ask the authors to change the wording of the text.
The explanation given for cultural competence is not correct. Cultural knowledge is one of the dimensions of this competence, but not the only one, so to say that because a nurse is culturally competent he or she can fall into stereotypes about different ethnicities is a misinterpretation. I suggest that the authors consult the abundant scientific literature that explores the concept of cultural competence from a holistic perspective, considering all its dimensions. I provide this bibliographical reference to facilitate consultation: Monteiro, A P., Melgar de Corral, G. Ugarte-Gurrutxaga, M. I. (Coords.) (2023). E-Book - MulticulturalCare Project: Training students through innovative learning methods to intervene in multicultural complex contexts (2020-1-PT01-KA203-078530). Coimbra School of Nursing; University of Castilla-La Mancha; UC Leuven-Limburg.
MATERIALS AND METHODS
In this section it is necessary to include missing information:
- It is necessary to make the profile of the sample visible, beyond saying that sex and age have participated. Even for this, it would be necessary to specify the age and sex of the person.
- It is necessary to include the script of the interview questions.
- It is necessary to include the analysis plan specifying the thematic axes.
RESULTS
The results are presented in a clear and orderly way, making them easy to read. However, there is one point that needs to be improved and that is the identification of each participant's discourse. It is necessary to include a code to know who is saying what is shown.
BIBLIOGRAPHY
We can see that there are references cited that are more than 8 years old. This is something that the authors should check, bearing in mind that the subject they are dealing with is very topical and there is a great deal of research dealing with it.
Author Response
Dear reviewer,
I thank you for the considerations indicated, which were essential for improving our manuscript. I'll show you below how they were integrated; I hope you can provide us with further considerations for improvement if necessary.
INTRODUCTION
The periphrasis has been rewritten and integrated with further references (lines 58-61).
The wording "non-citizens" has been changed (line 91)
We consulted the indicated reference and modified and integrated the information, which was ambiguous and incomplete; we hope we have managed to adequately clarify the concept (lines 158-169)
MATERIALS AND METHODS
We have clarified the characteristics of the sample better both through table 2 and by inserting a description in the "results" (lines 282-286)
We have included the interview guide in Table 1
We have included the data analysis with the thematic axes in Table 3.
RESULTS
We inserted the alphanumeric code with which the interviewees were identified after each quote.
BIBLIOGRAPHY
We are aware that several references are dated; however (unfortunately) in Italy this topic is not particularly investigated and, furthermore, other old studies on this topic carried out on Italian territory have shown that, despite the passing of the years and despite the legislative and organizational changes that (on paper) have been made , the barriers and obstacles that prevent migrants from receiving quality care are always the same.
We remain available for further changes.
Best regards.
Round 2
Reviewer 1 Report
Comments and Suggestions for Authors
Thank you for your sincere response to my review opinion. All responses or corrections made by the authors have been checked. I hope your studies prosper even further.
Thank you.
Reviewer 2 Report
Comments and Suggestions for Authors
Good afternoon,
After reviewing the changes made by the authors and taking into account the comments for improvement that I was able to provide, I consider the manuscript ready for publication.
Yours sincerely